# New Treatment Strategy Targeting Galectin-1 against Thyroid Cancer

**DOI:** 10.3390/cells10051112

**Published:** 2021-05-05

**Authors:** Laetitia Gheysen, Laura Soumoy, Anne Trelcat, Laurine Verset, Fabrice Journe, Sven Saussez

**Affiliations:** 1Laboratory of Human Anatomy and Experimental Oncology, Faculty of Medicine, Mons University, Avenue du Champ de Mars, 6, B7000 Mons, Belgium; laetitia.gheysen@umons.ac.be (L.G.); laura.soumoy@umons.ac.be (L.S.); anne.trelcat@umons.ac.be (A.T.); fabrice.journe@umons.ac.be (F.J.); 2Department of Pathology, Institut Jules Bordet, Université Libre de Bruxelles, 1000 Brussels, Belgium; laurine.verset@bordet.be; 3Laboratory of Clinical and Experimental Oncology, Institut Jules Bordet, Université Libre de Bruxelles, 1000 Brussels, Belgium; 4Department of Otorhinolaryngology and Head and Neck Surgery, CHU Saint-Pierre, Université Libre de Bruxelles, 1000 Brussels, Belgium

**Keywords:** OTX008, galectin 1, thyroid cancer, anaplastic thyroid cancer

## Abstract

Although the overall survival rate of papillary or follicular thyroid cancers is good, anaplastic carcinomas and radio iodine refractory cancers remain a significant therapeutic challenge. Galectin-1 (Gal-1) is overexpressed in tumor cells and tumor-associated endothelial cells, and is broadly implicated in angiogenesis, cancer cell motility and invasion, and immune system escape. Our team has previously demonstrated a higher serum level of Gal-1 in patients with differentiated thyroid cancers versus healthy patients, and explored, by a knockdown strategy, the effect of Gal-1 silencing on cell proliferation and invasion in vitro, and on tumor and metastasis development in vivo. OTX008 is a calixarene derivative designed to bind the Gal-1 amphipathic β-sheet conformation and has previously demonstrated anti-proliferative and anti-invasive properties in several cancer cell lines including colon, breast, head and neck, and prostate cancer lines. In the current work, the impacts of OTX008 were evaluated in six thyroid cancer cell lines, and significant inhibitions of proliferation, migration, and invasion were observed in all lines expressing high Gal-1 levels. In addition, the signaling pathways affected by this drug were examined using RPPA (reverse phase protein array) and phosphoprotein expression assays, and opposite regulation of eNos, PYK2, and HSP27 by OTX008 was detected by comparing the two anaplastic lines 8505c and CAL 62. Finally, the sensitive 8505c line was xenografted in nude mice, and 3 weeks of OTX008 treatment (5 mg/kg/day) demonstrated a significant reduction in tumor and lung metastasize sizes without side effects. Overall, OXT008 showed significant anti-cancer effects both in vitro and in vivo in thyroid cancer lines expressing Gal-1, supporting further investigation of the molecular mechanisms of the drug and future clinical trials in patients with anaplastic thyroid cancer.

## 1. Introduction

Thyroid cancer is the most common endocrine cancer and accounts for around 2% of total cancers diagnosed globally, corresponding to 567,233 new cases in 2018 and ranking in 9th place for incidence. The incidence of thyroid cancer has increased in many countries since the early 1980s, probably due to the increase in the diagnosis of papillary thyroid cancer in relation to the improvements in diagnostic methods [1]. Despite the fact that the overall survival rates of the main forms of thyroid cancer (papillary and follicular forms) are generally good, some forms, such as anaplastic carcinomas and radioactive iodine (RAI) refractory cancers, remain a significant therapeutic challenge [2,3]. Hence, study of the tumor microenvironment interactions, and understanding of drug resistance mechanisms, are crucial in the cancer field, particularly for thyroid anaplastic cancers, for which no therapy currently seems to improve the prognosis at one year.

Galectins represent a large family of lectins expressing a highly conserved sequence of approximately 130 amino acids called the “Carbohydrate Recognition Domain” (CRD). Galectins bind to a wide range of glycoproteins and glycolipids, both on the cell surface and in the extracellular matrix. By binding to these glycoconjugates, galectins transmit intracellular signals and mediate cell–cell and cell–extracellular matrix adhesion. Many studies have reported that galectins play crucial roles in cancer because they notably contribute to neoplastic transformation, tumor cell survival, angiogenesis, and tumor metastasis [4]. They also modulate the immune and inflammatory responses and might play a key role helping tumors to evade immune surveillance [4]. More specifically, galectin-1 (Gal-1) is reported as overexpressed in tumor cells and tumor-associated endothelial cells [5]. Gal-1 is broadly implicated in angiogenesis, cancer cell motility and invasion, and immune system escape [5]. An increase in its expression has been demonstrated in numerous solid cancers, including colorectal, lung, breast, pancreas, liver, and thyroid malignancies [6]. Importantly, intracellular Gal-1 interacts with the active form of the oncogenic H-Ras (H-Ras-GTP) and, therefore, increases its anchorage to the cell membrane, a crucial step in the malignant transformation [5]. In thyroid cancer, Gal-1 protein has been studied as a diagnostic marker and appears to be upregulated in papillary thyroid carcinoma and its lymph node metastases [7]. The role of Gal-1 in thyroid cancer has been previously explored by our team, who reported a higher serum level of Gal-1 in patients with differentiated thyroid cancer than in healthy patients [8]. We also demonstrated the implication of Gal-1 in thyroid cancer by using a knockdown model showing the inhibition of TPC-1 cell migration, 8505c cell proliferation, and invasion in vitro, in addition to a significant decrease in tumor and metastasis development in vivo [9]. Overall, these data indicate that Gal-1 could represent an interesting therapeutic target in thyroid cancers.

Although antibodies and peptides may be an interesting option to inhibit Gal-1, small molecules tend to be advantageous regarding bioavailability, immunogenic activation, degradation, and ability to scale up [10]. OTX008 is a calixarene derivative designed to bind the Gal-1 amphipathic β-sheet conformation [11]. Nuclear magnetic resonance analysis demonstrated that OTX008 targets Gal-1 at a site distant from the lectin CRD and acts as an allosteric inhibitor of glycan/carbohydrate binding [12]. This molecule has previously shown anti-proliferative and anti-invasive properties at micromolar concentrations in colon, breast, head and neck, prostate, ovarian, renal, and lung cancer cell lines [11]. Results in an ovarian xenograft model confirmed the anti-proliferative effects observed in vitro, in addition to reduction in the micro-vessel diameter and the inhibition of VEGFR2 expression within the tumor [11].

Our study aimed to evaluate the anti-tumoral effects of OTX008 in a panel of six thyroid cancer cell lines, in addition to the use of a xenograft mouse model for anaplastic cancer. Moreover, we addressed the molecular mechanisms of action of OTX008 in our anaplastic model by conducting RPPA and phosphoprotein expression assays.

## 2. Materials and Methods

### 2.1. Cell Lines

Human thyroid cancer cell lines 8505c (ATC), BCPAP (PTC), TPC-1 (PTC) were kindly provided by Prof. C. Maenhaut, (IRIBHM, Université Libre de Bruxelles, Brussels, Belgium). The FTC-133 (FTC) lines were kindly gifted by Dr. Köhrke (Institute of Experimental Endocrinology of the Charité, Humbold University, Berlin, Germany) and the TT2609C02 (FTC) and CAL-62 (ATC) cell lines were purchased from DSMZ company (Braunschweig, Germany). All cell lines except CAL-62 were maintained in RPMI-1640 with L-Glutamine (Lonza, Westburg, The Netherlands) supplemented with 10% fetal bovine serum, Brazil (Gibco™, Thermo Fisher, Bruxelles Belgium) and 1% penicillin/streptomycin (Gibco™, Thermo Fisher, Bruxelles, Belgium). The CAL-62 cell line was cultured in Dulbecco’s Modified Eagle Medium (DMEM 4.5 g/L glucose without L-Glutamine, Lonza, Westburg, The Netherlands) supplemented with 10% fetal bovine serum (Gibco™, Thermo Fisher, Bruxelles, Belgium), 2% glutamine (Gibco™, Thermo Fisher, Bruxelles, Belgium) and 1% penicillin/streptomycin (Gibco™, Thermo Fisher, Bruxelles, Belgium). All cell lines were cultured in a standard humidified incubator at 37 °C in 5% CO_2_ atmosphere and monthly tested for Mycoplasma by PCR test. The culture medium was changed twice a week. For routine maintenance and experimental studies, cells were detached by incubation with accutase solution (Gibco™, Thermo Fisher, Bruxelles Belgium), resuspended, and counted using an electronic cell counter (Scepter Cell Counter Sensors, Millipore, Merck, Overijse, Belgium).

### 2.2. Cell Proliferation Assay

Cell proliferation assays were performed in 96 well plates (Sarstedt, Berchem, Belgium) in triplicate. The 8505c, TPC-1, FTC133, and BCPAP were plated at 1500 cells/well in 100 µL of culture media, whereas the TT2609C02 were plated at 3000 cells/well and CAL-62 at 2000 cells/well. The next day, cells were supplemented with 100 µL of fresh medium containing OTX008 drug (HY-19756, MedChemExpress, Bio-Connect BV, Te Huissen, The Netherlands) at gradual concentrations or with vehicle for control. Medium with drug was removed after 72 h and cells were washed with DPBS supplemented with Mg++ and Ca++ (Lonza, Westburg, The Netherlands) and then fixed with a solution of 1% glutaraldehyde (Sigma-Aldrich, Merck, Overijse, Belgium) for 15 min. After washing, the cells were colored by a solution of 4% crystal violet (Sigma-Aldrich, Merck, Overijse, Belgium) for 30 min. The cells were washed with tap water and 96 well plates were left for a minimum of 1 h on a bench in open air to dry. Cells were finally permeabilized with a solution of Triton™ X-100 (Sigma-Aldrich, Merck, Overijse, Belgium) for 90 min. Finally, the absorbance was determined at 570 nm with a spectrophotometer (VERSA max-SoftMax Pro, VWR, Leuven, Belgium) and the results were normalized to control (untreated cells).

### 2.3. Clonogenic Assay

Cells (250 cells/well for each cell line) were seeded in 6 well plates (Nunc™, VWR, Leuven, Belgium) and allowed to adhere for 48 h in culture medium. The cells were then exposed to OTX008 at gradual concentrations or with vehicle for control. The medium was refreshed once during the 10 days of growth culture. At day 10, the medium was removed, and cells were washed with DPBS supplemented with Ca++ and Mg++ (Lonza, Westburg, The Netherlands). The cells were then fixed with 1% glutaraldehyde (Sigma-Aldrich, Overijse, Belgium) for 15 min and then stained with 4% crystal violet solution (Sigma-Aldrich, Overijse, Belgium) for 30 min. The excess of crystal violet was removed with tap water and the colonies were counted.

### 2.4. Cell Migration and Invasion Assay

The migration and invasion assay protocols were similar and briefly described below. The cell kits were purchased from Merck Company (QCM Chemotaxis Cell Migration Assay, 24 well, 8 µm and QCM ECMatrix Cell Invasion Assay, 24 well, 8 µm, Merck, Overijse, Belgium), and presented an 8 µm pore size polycarbonate membrane, alone (for migration assessment) or over which a thin layer of ECMatrix™ is dried (for invasion evaluation). Firstly, the chambers were hydrated with medium without FBS during minimum 30 min in the cell incubator. Medium supplemented with FBS was added in each well. Cells were suspended in medium without FBS at line-dependent densities. The cell suspension was placed in the center of the chamber. An equal volume of solution of medium containing OTX008 was added in cell suspension depending on the tested conditions. Plates were then incubated for 48 h at 37 °C. At the end of treatment, the cells inside the chambers were removed, and excess cells were eliminated with a cotton swab. The external bottom part of the chamber, where cells migrated or invaded, was fixed with a solution of 1% glutaraldehyde for 15 min. A solution of 4% crystal violet was added in wells, whereas chambers were deposited on it and incubated for 2 min. Finally, chambers were washed with distilled water. Cell invasion was observed under microscope. Quantification was calculated after 1 h solubilization in Triton™ X-100 solution and the relative absorbance was determined at 570 nm.

### 2.5. Protein Microarray

Protein microarray was performed by RPPA analysis at the Functional Proteomics Core Facility at The University of Texas MD Anderson Cancer Center. The cell extracts were sent to the University of Texas. Cells were plated in T75 and detached by incubation with accutase solution (Gibco™, Thermo Fisher, Bruxelles, Belgium). The second step was counting using an electronic cell counter (Scepter Cell Counter Sensors, Millipore, Merck, Overijse Belgium), followed by centrifugation at 4 °C, 1200 rpm for 5 min. Successive washes were performed to finally transfer 6 × 10^6^ cells in an Eppendorf. Cells were centrifuged and stored at −80 °C.

### 2.6. Human Phospho-Kinase Array

A Human Phospho-Kinase Array kit (R&D Systems, Minneapolis, MN, USA) was used to evaluate variations of 37 kinase phosphorylation sites after OTX treatment. First, 8505c and CAL-62 cells were plated in dishes for 72 h. Once the cells were at confluence in dishes, they were treated with 30 µM OTX008 for 30 min. Media was then eliminated, and cells were washed with DPBS with Ca++ et Mg++ on ice and finally scratched in DPBS. Cells were centrifuged at 1200 rpm for 5 min at 4 °C and the pellets were solubilized in lysis buffer supplemented with phosphatase provided in the Human Phospho Array kit. The lysis buffer was incubated for 30 min at 4 °C. Cell suspension was finally centrifuged at 14,000 rpm for 5 min at 4 °C. The supernatant was transferred and stored at −80 °C. Quantitation of sample protein concentration was performed using BCA Protein Assay (Pierce™ BCA Protein Assay Kit, Thermo Scientific™, Waltham, MA, USA) using bovine serum albumin as standard. The nitrocellulose membranes from the kit were blocked with provided solution and the cell lysates were incubated with the membranes overnight at 4 °C on a rocking platform shaker. After 3 washings, the cocktails of antibodies were incubated with the membranes for 2 h at room temperature on a rocking platform. The next steps comprised 3 washings followed by the incubation with the streptavidin-HRP for 30 min at room temperature on a rocking platform. After a final step of washing, the chemi-reagent mix was added, and the spot revealed with a LAS-3000 CCD camera (Fujifilm, Tokyo, Japan), using software specifically designed for image acquisition (Image Reader, Raytest^®^, Straubenhardt, Germany). Immunoreactive spot intensities were quantified using the software AIDA^®^ Image Analyser 3.45 (Raytest^®^, Straubenhardt, Germany).

### 2.7. Immunofluorescence Assay

Cells were plated at day 0 in 12 well plates with 1 mL of medium per well. Each well contained a coverslip. Cell densities varied according to conditions (no treatment: 3000 cells/well, 10 µM OTX008: 10,000 cells/well, 30 µM OTX008: 30 000 cells/well). At day 1, drug or vehicle was added to the medium for 72 h. The steps of fixation and antibody incubation were performed at day 4. Cells were washed with PBS and then fixed for 15 min in paraformaldehyde solution. Cells were washed again with a PBS solution containing 0.1% Triton™ X-100 for 3 successive washes of 5 min. A blocking solution of PBS-BSA 5% was added for 1 h at room temperature. Cells were again washed with PBS solution. The primary antibody (Galectin 1 Monoclonal Antibody (6C8.4-1) Thermo Fisher Scientific, Waltham, MA, USA) was added at a dilution of 1:250 in PBS containing 0.5% BSA for 3 h at room temperature (negative control lacked antibodies). Cells were washed 3 times for 5 min with 0.1% Triton™ X-100 in PBS. The secondary antibody (Alexa fluor) 1:100 in PBS containing 0.5% BSA was then added for 30 min at room temperature. Cells were finally washed 3 times with 0.1% Triton™ X-100 in PBS. A Few drops of DAPI solution (ProLong™ Gold Antifade Mountant with DAPI, Thermo Fisher Scientific, Waltham, MA, USA) were added to a glass side, which was then covered by a coverslip. Images were then acquired with NIS element viewer 5.21.

### 2.8. pERK Detection by HTFR Kit

The 8505c cells were plated in 96 well plates 24 h before the test. The cells were then treated with 30 µM OTX008 treatment for 10 and 30 min. A lysis buffer provided in the kit was added to the cells and incubated during 45 min on a rocking platform. The lysates were then transferred to the provided plate and the antibody added. An incubation of 4 h at room temperature was performed. The plate was finally read at 665 and 620 nm with the Spectra Max5. 

### 2.9. Xenograft ATC Models

Four to nine week old male athymic nude mice were purchased from Charles River (490CRATHHO, Male CR ATH HO MOUSE 28–34 days, Charles River, Sait Germain Nuelles, France), which were fed ad libitum and kept in optimal conditions in a 12 h light/dark cycle. Upon arrival, animals were isolated in the animal facility for a 1 week quarantine before starting experiments. The 8505c cells (1 × 10^6^) were resuspended in 300 μL HBSS (no calcium, no magnesium, no phenol red, Gibco™, Thermo Fisher, Bruxelles, Belgium) and injected subcutaneously on the right flank of the immunodeficient mice. A total of 9 mice were inoculated with cancer cells (3 in the control group and 6 in the drug-treated group). The treatment was administered to mice by daily intraperitoneal (IP) injections of either vehicle or OTX008 at 5 mg/kg/day for 5 days/week, 2 days off, for 3 weeks. The mice weight was measured every two days. The tumor development was monitored 3 times/week and tumor volumes were evaluated by measurements of tumor length and width in millimeters using calipers. At the end of treatment, the animals were sacrificed by lethal IP injection of Dolethal 200 mg/mL (pentobarbital sodique, Vetoquinol, Aartselaar, Belgium). Tumor tissues and lungs were collected. Tumor volumes were calculated according to the formula (length × width × thickness)/2. Tissues were fixed in 4% formaldehyde and paraffined for HE staining and histological analyses. All animal experiments were performed according to the institutional guidelines and approved by the ethics committee of the University of Mons (Mons, Belgium) (SA-04-01).

### 2.10. Statistical Analysis

All statistical analyses were performed using IBM SPSS Statistics 23 (IBM, Ehningen, Germany). In vitro data are expressed as means ± standard deviation, and comparison of mean values was conducted with one way ANOVA test and Tukey (multiple variables compared each other) or Dunnett (multiple variables compared to control) post hoc tests, or with the *t*-test (comparison of two variables). Animal data are presented in box plots (median and percentiles 5/25/75/95) and analyzed by the Mann Whitney test. Significance was indicated as * *p* < 0.05; ** *p* < 0.01; *** *p* < 0.001.

## 3. Results

### 3.1. Effect of OTX008 on Thyroid Cancer Cell Proliferation

To evaluate the effect of Gal-1 inhibition by OTX008 on cell proliferation, six thyroid cancer cell lines (two papillary lines—TPC1, BCPAP; two follicular lines—FTC133, TT2609C02; and two anaplastic lines—8505c, CAL62) were cultured with a large range of OTX008 concentrations for 72 h and stained with crystal violet to evaluate cell proliferation. Results are presented in Figure 1, where each graph combines two lines of the papillary, follicular, and anaplastic forms. The concentration–response curves showed similar profiles for all lines, except for the CAL62 line, which appeared to be significantly more resistant to the inhibitor. These curves allow assessment of the IC50 values defined as the OTX008 concentrations able to inhibit cell proliferation by 50%. The IC50 values for each cell line are presented in Table 1; these varied between 0.2 and 1 µM in sensitive lines, and the value for the resistant CAL62 line was estimated to be 30 µM, using the GraphPad Prism software based on cell proliferation data.

### 3.2. Effect of OTX008 on Colony Formation in Thyroid Cancer Cells

In this experiment, we assessed the long-term effect (10 days) of OTX008 by performing colony formation assays in five thyroid cancer lines (TPC1, BCPAP, FTC133, 8505c, CAL62). Of note, long-term culture was not possible for the TT2609C02 line because adhesion of these cells was too low. After treatment, cells were stained by crystal violet to visualize cell colonies and evaluate the impact of OTX008 on their formation and size. Figure 2 presents the results of the clonogenic assays and shows that colony formation was highly inhibited by 10^−6^ M OTX008 in all cell lines, except in the CAL62 line, which was largely unaffected by the drug. Of note, colony formation with CAL62 cells was very high, reflecting the aggressiveness of this line.

### 3.3. Effect of OTX008 on Cell Migration in Thyroid Cancer Lines

The ability of migration of the six thyroid cancer cell lines was evaluated using Boyden chambers. Cells were exposed to 1 or 3 µM OTX008 for 48 h (Figure 3A). OTX008 concentrations were chosen based on Astorgues-Xerri experiments and then adapted to our cell lines [11]. Only the most relevant concentrations were tested, and colorimetric analyses were performed using crystal violet staining (Figure 3B). The two lines, CAL-62 and TT2609C02, were only slightly impacted by 3 µM OTX008 and, respectively, 91% and 82% of cells were still able to migrate. In comparison, 3 µM OTX008 induced a moderate effect (56% residual migration) in the 8505c line, and a stronger inhibition (from 37% to 25% residual migration) in the three other lines (TPC-1, FTC133, and BCPAP). These latter lines already exhibited about 50% of inhibition of cell migration at 1 µM OTX008.

### 3.4. Effect of OTX008 on Cell Invasion in Thyroid Cancer Lines

For invasion assays, cells were exposed to 3 µM OTX008 for 48 h (Figure 4A) in the inserts of Boyden chambers containing ECMatrix™, a reconstituted basement membrane matrix of proteins derived from the Engelbreth Holm-Swarm (EHS) mouse tumor. Regarding migration assays, CAL-62 cells showed the weakest inhibition of invasion after treatment with OTX008 (Figure 4B). A moderate effect was notable in FTC133, TPC-1, and BCPAP cells with, respectively, 52%, 58%, and 62% of cells still able to invade. A greater impact of OTX008 was observed in 8505c and TT2609C02 cells with, respectively, 41% and 45% of invading cells showing the high capacity of the Gal-1 inhibitor to reduce the invasiveness of these latter cells.

### 3.5. RPPA Analyses Show Different Phosphorylation and Protein Profiles in Relation with Sensitivity or Resistance of Thyroid Cancer Cell Lines to OTX008

Using a Reverse Phase Protein Array (RPPA) strategy (University of Texas, MD Anderson Cancer Center), we obtained the rule data from a large-scale screening of phosphorylation and protein levels in our six thyroid cancer cell lines. The RPPA data were correlated to the IC50 values for the inhibition of proliferation of cells exposed to OTX008 (Spearman rho correlation). This analysis was conducted to classify basal phosphorylation and protein levels based on the sensitivity or resistance of the cells to OTX008. Regarding the protein profile in CAL-62, two groups of proteins were identified, representing up-regulated (*n* = 35) and down-regulated (*n* = 17) proteins in this line. No relationship associated with the thyroid cancer form (papillary, follicular, or anaplastic) was identified. Indeed, heatmaps in Figure 5 show that the lines 8505c and CAL62, both anaplastic, have opposite phosphorylation levels for a large number of proteins. By contrast, both CAL62 and TT2609C02 lines share similar protein profiles for the up-regulated protein group, whereas this is not the case for the down-regulated proteins. The proteins highlighted by this analysis are part of multiple signaling pathways involved in cell proliferation (MAPK8, PDGFR, RPS6KA1, EIF4EBP1, PTK2, ERBB2, ERBB3, RPS6KA1/2/3, SGK3, AKT1, PIK3CA), angiogenesis and migration (EPHA2, SERPINE1), iron metabolism (TFRC), immunity (LCK, CD276, PTK2, YES1, ANXA1, GATA3), apoptosis (ABL1, BBC3, BIRC3, FOXO3, YAP1), and DNA damage and repair (DDB1, HES1, MYH11). Such phosphorylations and protein profiles provide new insight into the thyroid cancer lines and identify, in CAL 62 cells, potential proteins involved in resistance to OTX008, such as HSP27, eNOS, and PYK2.

### 3.6. Effect of OTX008 on Protein Phosphorylation in Anaplastic Cells According to the RAS Mutation Status

To better understand the molecular mechanisms of action of OTX008, the sensitive cell line (8505c, WT RAS) and the resistant line (CAL 62, mutant KRAS) were compared after 30 min of treatment with 30 µM OTX008 through the evaluation of the phosphorylation prolife of 37 phosphokinases using the Human Phospho-Kinase Array kit from R&D Systems. The analyses of the phosphorylation rates, comparing treated and untreated conditions, revealed some disparities between CAL 62 and 8505c lines. In the 8505c cell line, most of the proteins showed weak variations in term of phosphorylation, as presented in Figure 6. However, a strong increase in phosphorylation was observed for STAT3 and STAT1, whereas a drastic decrease was reported for HSP27 and WNK1. In the case of CAL 62 cells, a high number of proteins showed changes in phosphorylation rate following OTX008 treatment. Most of these variations were decreases of phosphorylation, as exemplified by eNOS, whereas important increases were observed for HSP27 and PYK2. Comparing the phosphorylation levels in both cell lines, HSP27 protein was clearly differentially expressed with a down-regulation in 8505c and an up-regulation in the CAL 62 cells. To a lesser extent, eNOS showed the inverse variation, with an up-regulation in 8505c and a down-regulation in CAL 62 cells. Of note, the phosphorylation level of the WNK1 protein appeared to trend down in both cell lines. Overall, OTX008 treatment induced a more important variation in the phosphorylation of some proteins in the CAL 62 line, which could be due to the presence of a mutation in KRAS. In addition, HSP27 and eNOS, already sorted from RPPA analyses, were regulated in the opposite way in the two anaplastic lines, which could be linked to the difference in sensitivity of these cells to the drug.

### 3.7. Effect of OTX008 on the Expression and the Subcellular Localization of Gal-1 in Anaplastic Cell Lines

The expression of Gal-1 has been evaluated by immunofluorescence and confocal microscopy in the six lines (Appendix A). In all cell lines, Gal-1 was detected in both cytoplasm and nucleus. The highest expression was found in the 8505C cells while the lowest level was in the CAL-62 cells. More specifically, Gal-1 was detected in both cytoplasm and nuclear compartments in 8505c cells while it was mainly observed in the cytoplasm of the CAL 62 line. Because these two lines had opposite responses to OTX008 and opposite Gal-1 expressions, these lines were submitted to OTX treatment during 72 h in different conditions (CTR, 10 and 30 µM OTX008) in order to assess Gal-1 levels (Figure 7). In 8505c cells, treatment for 72 h with 10 µM OTX008 showed minor effects on Gal-1 expression and localization, whereas 30 µM OTX008 induced a strong decrease of Gal-1 in the cytoplasm. In CAL 62 cells, exposure to 10 µM OTX008 had no impact on Gal-1 expression and localization, but 30 µM OTX008 resulted in an important increase of Gal-1 in the cytoplasm and nucleus, and in association with a modification of the cell morphology.

### 3.8. OTX008 Effects on ERK Phosphorylation in 8505c Anaplastic Thyroid Cancer Cells

The ability of Gal-1 to bind RAS is well documented in the literature. To confirm the inhibition of RAS by OTX008 treatment in the 8505c cell line and the subsequent MAPK signaling pathway, we used a pERK HTRF kit and demonstrated a weak inhibition (about 10%) of pERK in 8505c cells treated with 10 µM OTX008 for 10 min. A drastic inhibition was observed after 30 min of treatment with approximately 65% of phosphorylation inhibition (Figure 8), indicating an important inhibitory effect of OTX on the MAPK signaling pathway.

### 3.9. Effect of OTX008 on Anaplastic Tumor Growth in Xenograft Mouse Model

The efficiency of OTX008 on the growth of anaplastic tumors was next evaluated using a xenograft mouse model. A suspension of 8505c cells was subcutaneously injected in the right flank of nude mice. When tumor development became visible and measurable (10 days post-injection of the cells), the treatment (vehicle or 5 mg/kg OTX008) was administered daily, 5 days per week, 2 days off, for 3 weeks. As presented in Figure 9, OTX008 significantly decreased tumor volume when it was evaluated after 3 weeks of treatment. The weight and behavior of each animal was monitored daily and in no cases were signs of distress or toxicity linked to the drug observed. In addition, macroscopic observations of the various organs at autopsy did not reveal any abnormalities.

### 3.10. Effect of OTX008 on Metastasis Development in Mice

The development of lung metastasis was evaluated in the two groups of mice (no treated (control) vs. 5 mg/kg OTX008). After 3 weeks of treatment and euthanasia of animals, lungs were collected, fixed in 4% formaldehyde, and conserved in paraffine. Five 4 µm tissue sections were cut each 40 µM to examine a large part of each lung and were colored with hematoxylin/eosine solution for histological evaluation. The largest size of each metastasis was measured for both experimental groups and results are presented in Figure 10. OTX008 treatment significantly reduced the size of lung metastases.

## 4. Discussion

The aim of our study was to deepen our knowledge on the importance of Gal-1 in the biology of thyroid cancer. Due to the effect of the OTX008 molecule, we observed significant inhibition of cell proliferation and invasion in vitro, and of tumor and metastasis growth in a xenografted mouse model. Experiments were also conducted to further understand the molecular mechanisms involved in the anti-cancer properties of OTX008 in thyroid cancer. OTX008, which is a small molecule acting as an allosteric inhibitor of Gal-1, appears to present therapeutic potential in cancer. OTX008 is chemically more stable and resistant to hydrolysis in comparison to other Gal-1 inhibitors such as Anginex. Its low molecular weight (937 Da), associated with the fact that it is neither a protein or a saccharide, but a phenyl-based molecule, greatly increases its bioavailability [6].

First, our in vitro study showed that OTX008 exhibits direct anti-tumor activity in vitro due to an anti-proliferative effect on five different thyroid cancer cell lines (8505c, TPC1, BCPAP, FTC133, TT2609C02), with IC50 values ranging from 0.2 to 1.1 µM. By contrast, the anaplastic thyroid cancer cell CAL62 showed a limited response to the drug, with an IC50 value > 30 µM. The anti-proliferative properties of OTX008 were previously demonstrated in many cancer cell lines, including colon (HT29, COLO205-S, HCC2998, HCT116, COLO205-R), ovarian (A2780-1A9, OVCAR 3, IGROV1, SKOV3), prostate (PC3, DU 145), head and neck (SQ20B, SCC91, HEP2), breast (MCFS7, SK BR3, ZR-75-1, MCF7-shWISP), and renal cancer lines (CAKI 1), with IC50 values ranging from 3 to 500 µM, demonstrating the variable response which can be obtained with such a Gal-1 inhibitor [11]. Variable results were also obtained for melanoma (WM983a, A375 M) and glioblastoma (U87MG), with IC50 values between 35 µM to 192 µM. This variability observed between all cell lines underlines the possibility of intrinsic molecular mechanisms for OTX008 action, or specific targets for OTX008 depending on cell lines, driving the response to the drug.

Then, an in vivo model was developed based on our in vitro results. This model reported a significant reduction in tumor growth, to 66% of that of the control group, and inhibition of lung metastasis development in mice treated with 5 mg/kg OTX008 for 3 weeks. In this context, Xerri’s study reported that OTX008 induced a significant reduction in tumor growth in nude mice bearing A2780-1A9 ovarian cancer xenografts after 2 weeks of treatment [11]. OTX008 has also been shown to be efficient in two other models of carcinoma using MA148 human ovarian and B16 mouse melanoma by subcutaneous delivery of the drug with implanted osmotic mini pumps. In the MA148 model, ovarian tumor growth was reduced to 58% at 2.4 mg/kg [13]. In B16 mouse melanoma, the inhibition of tumor growth reached 77% at 10 mg/kg in comparison to the untreated group control [13]. Of note, at these concentrations, the authors did not observe toxicity, as assessed by animal behavior, body weight change, or hematocrit or creatinine levels (both determined by blood drawing on the last day of treatment). Moreover, after autopsy, no readily apparent abnormalities in internal organs were observed [13]. In our study, the in vivo evaluation of OTX008 in an anaplastic thyroid carcinoma xenograft model also resulted in the absence of notable side-effects.

The aspects of invasiveness and migration studied in our work showed that OTX008 inhibits the migrative property in four thyroid cell lines (8505c, FTC133, TPC1, BCPAP), with more than 50% of inhibition observed with 3 µM treatment after 48 h. In Xerri’s study, OTX008 demonstrated its capacity to inhibit the invasion of SQ20B cells at micromolar concentrations. In addition, Zuchetti and colleagues demonstrated an inhibition, almost to the baseline level, of the endothelial cell motility and invasion after OTX008 treatment [14]. Overall, these results confirm the anti-invasive and anti-migrative properties of OTX008 in different cell types.

A hypothesis to explain the sensitivity/resistance of thyroid cancer cells to OTX008 is the activation of cell-dependent molecular mechanisms. Based on our in vitro data, we explored the impact of OTX008 on various signaling pathways, Gal-1 expression, and subcellular localization, with regard to RAS mutation. Many studies have reported that galectins drive an intracellular signaling pathway via interaction with the small GTPase RAS [15,16,17,18]. In our study, the analysis of the RAS mutation in our cell lines revealed that only the CAL-62 line harbors a mutation in KRAS, with probably fewer interactions with Gal-1, supporting the lack of efficacy of OTX008 in this cell line.

In addition, given our results from phospho-kinase array and the literature, a cascade implicating Gal-1, integrin β, PYK2, eNOS, and HSP27 is proposed to explain the opposite response to OTX008 in CAL-62 and 8505c. Indeed, we evaluated the phosphorylation levels of many proteins and noticed some important disparities between the two lines, such as an inverse expression of HSP27 S78/S82 and eNOS S1177. Highly expressed in CAL-62, HSP27 is known as a marker of resistance and is related to the overall survival rate, notably in many cancers, such as breast cancer, prostate cancer, gastric cancer, lung cancer or liver cancer [19,20,21,22]. Therefore, our results appear to confirm the role of HSP27 in the resistance of the anaplastic thyroid cancer to OTX008. In contrast, our study demonstrated that the level of S1177 phosphorylation of eNOS was significantly lower in CAL-62 than in 8505c lines, and the final major disparity related to the Y402 phosphorylation of PYK2 (proline-rich tyrosine kinase 2), with a higher expression in CAL-62 than in 8505c cells. Overall, CAL-62 resistance is characterized by high phosphorylation of HSP27 and PYK2, and weak phosphorylation of eNOS.

Of note, PYK2 is part of the focal adhesion kinase (FAK) non-receptor tyrosine kinase family and can be activated by multiple growth factors (GFs), hormones, neuropeptides, cytokines, and chemokines [23]. PYK2 is implicated in many signal transduction cascades and plays critical roles in controlling cell adhesion, proliferation, migration, and invasion. Most importantly, recent studies suggest a network between PYK2, eNOS, and HSP27. In breast cancer, the strong phosphorylation levels of HSP27 and PYK2 have been associated with a form of resistance to doxorubicin [24]. In addition, the inhibition of eNOS mediated by PYK2 has been explored in cardiomyocyte survival and cardio-protection, showing a possible cascade between these molecules [25].

As previously mentioned, OTX008 acts as an inhibitor of Gal-1, which is broadly implicated in cell–cell and cell–extracellular matrix interactions, and its implication in tumor invasion has been broadly described in the literature [4,5,6]. The integrin β1 seems to exert an important connection between Gal-1 and the FAK signaling pathway. By a knockdown of Gal-1 in breast cancer, the Gal-1/integrin β1 interactions have been highlighted because the depletion of Gal-1 negatively impacts the signaling pathway downstream of FAK/SRC [26]. Another study evaluated the impact of Gal-1 overexpression and demonstrated the activation of the FAK/PI3K/AKT/mTOR pathway, which is correlated with upper urinary urothelial carcinoma progression and survival [27]. The characterization of integrin receptor molecules in different thyroid cancer cell lines confirmed high levels of integrin β1 in anaplastic thyroid cancers [28]. Integrin linked kinase has also been studied, showing an elevated expression in anaplastic thyroid cancer [28,29,30]. All of these arguments highlighting the relationship between Gal-1, integrin (more specifically, integrin β1), and the FAK pathway support our hypothesis of a relationship between Gal-1 and PYK2, member of the FAK family. To confirm the possible activation of PYK2 through an interaction between Gal-1 and integrin β1, it would be interesting to assess the effect of PYK2 inhibitors on the phosphorylation levels of eNOS and HSP27.

## 5. Conclusions

Our study demonstrated the therapeutic potential of OTX008 in thyroid cancer cells based on the inhibition of proliferation and invasion in vitro, in addition to reduction in tumor mass and the development of lung metastasis in a xenograft anaplastic mouse model. However, resistance to OTX008 can be observed in the KRAS mutated line, suggesting crosstalk between Gal-1 and KRAS for signal transduction in thyroid cancer cells. In addition, our data support a possible mechanism of resistance through the activation of a cascade involving Gal-1, integrin β1, PYK2, eNOS, and HSP27. More investigations are nevertheless necessary to confirm this hypothesis and optimize the treatment of the promising molecule OTX008 in combination with inhibitors of such intracellular cascade.

## Figures and Tables

**Figure 1 cells-10-01112-f001:**
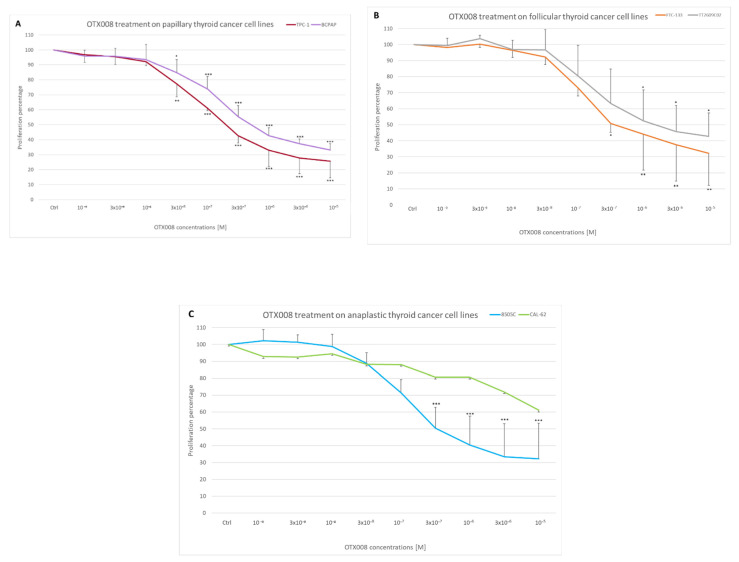
Effect of OTX008 treatment on thyroid cancer cell proliferation. Cells were cultured in 96 well plates and exposed to various concentrations of OTX008 (10^−9^–10^−5^ M) for 72 h. Cells were fixed in PAF, stained with crystal violet, solubilized with Triton X-100, and the OD of staining was determined at 570 nm. Data present the concentration–response curves of (**A**) papillary thyroid cancer cell lines, (**B**) follicular thyroid cancer lines, and (**C**) anaplastic lines. Control values were normalized at 100% and compared to treatment values. ANOVA and Dunnett post hoc tests were performed to assess significance (* *p* < 0.05; ** *p* < 0.01; *** *p* < 0.001).

**Figure 2 cells-10-01112-f002:**
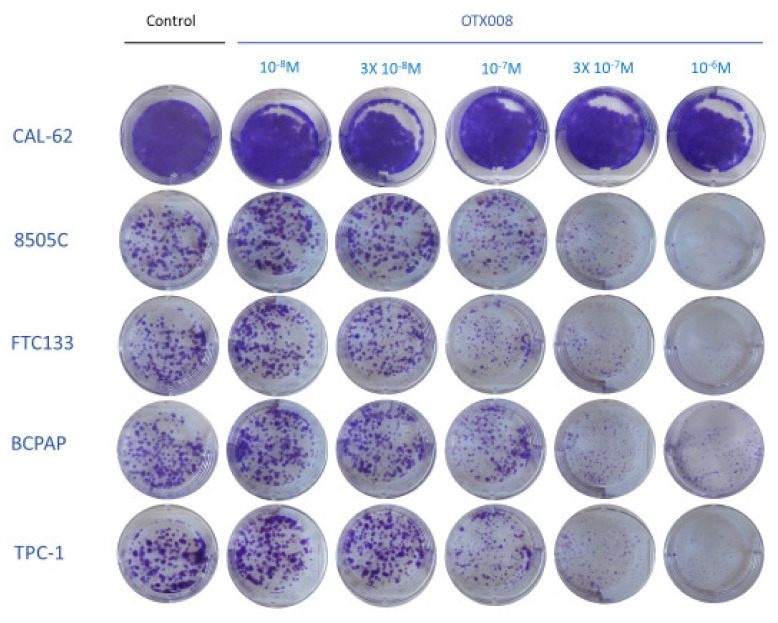
Evaluation of long-term effect of OTX008 on colony formation of thyroid cancer cells. Cells were treated with OTX008 concentrations ranging from 10^−8^ to 10^−6^ M for 10 days in six well plates, then fixed and stained with crystal violet solution to visualize cell colonies.

**Figure 3 cells-10-01112-f003:**
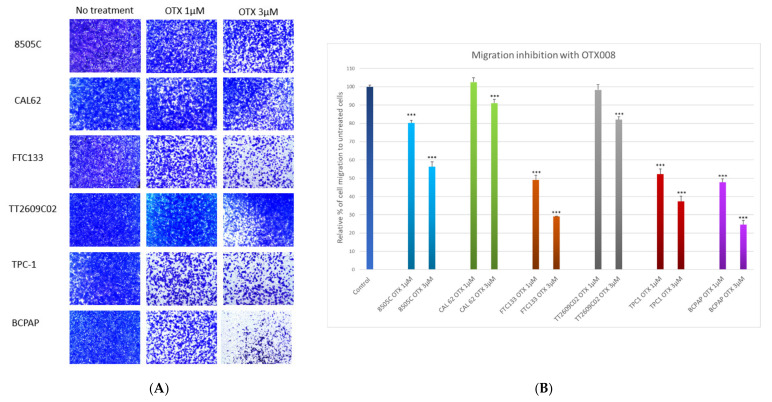
Evaluation of the anti-migrative properties of OTX008 in thyroid cancer cell lines. (**A**) Cells were plated in a Boyden chamber in media without FBS and with 1 or 3 µM OTX008 for 48 h. (**B**) The percentage of invasiveness was determined by solubilization with Triton X-100 of invasive cells stained by crystal violet. The quantification was performed by measuring the optical density at 570 nm. Migration values for untreated cells are normalized at 100% for each line and are represented once as Control. Statistical analyses refer to each respective control line. ANOVA and Dunnett post hoc tests were conducted to assess significance compared to respective control cells whose migrations were normalized at 100% (*** *p* < 0.001).

**Figure 4 cells-10-01112-f004:**
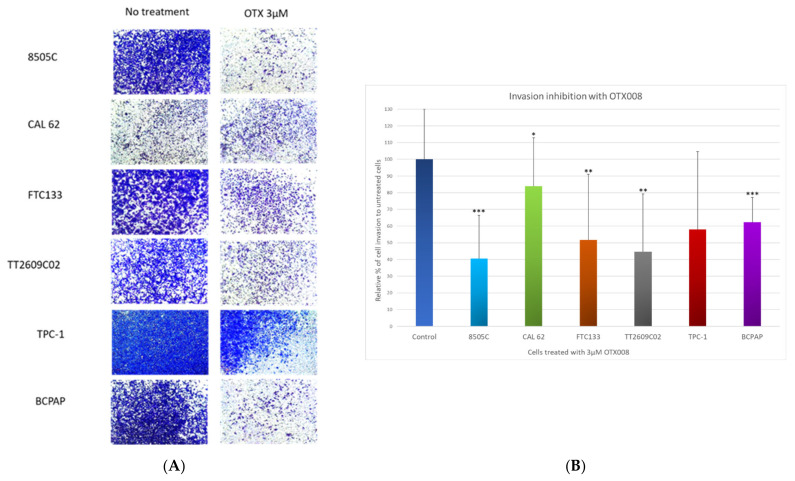
Evaluation of the anti-invasive properties of OTX008 in thyroid cancer cell lines. (**A**) Cells were plated in a Boyden chamber in media without FBS and with 3 µM OTX008 for 48 h. (**B**) The percentage of invasiveness was determined by solubilization of invasive cells and quantification of the crystal violet staining at the optical density of 570 nm. Invasion values for untreated cells are normalized at 100% for each line and are represented once as Control. Statistical analyses refer to each respective control line. Student’s t-test was performed to assess significance regarding to respective control cells whose invasion capacity were normalized at 100% (* *p* < 0.05; ** *p* < 0.01; *** *p* < 0.001).

**Figure 5 cells-10-01112-f005:**
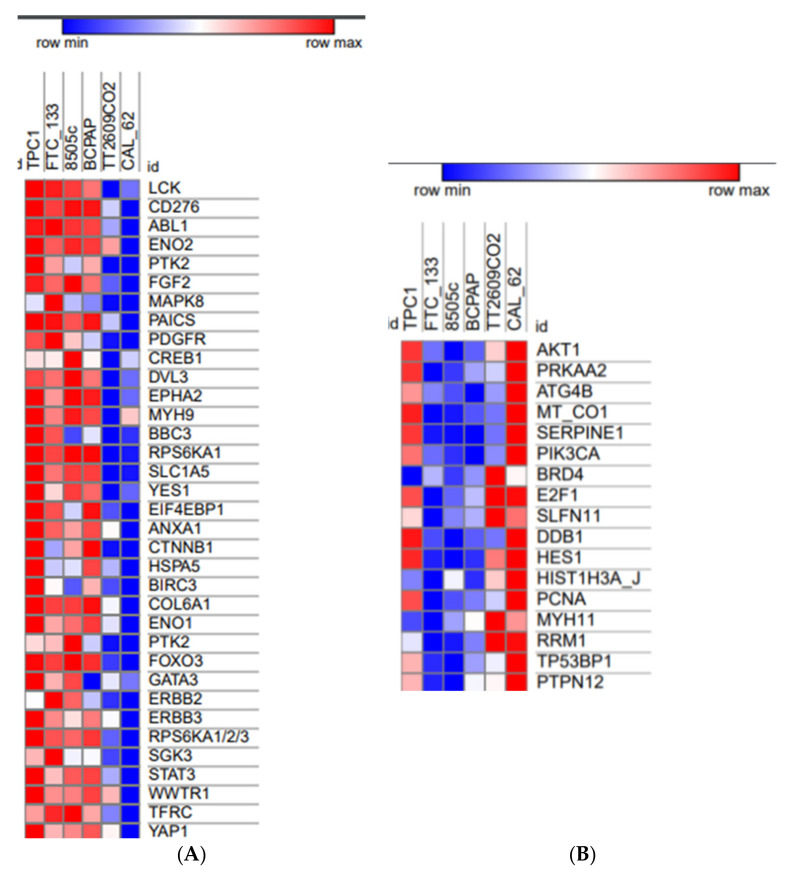
Heatmaps generated from RPPA data comparing thyroid cell lines based on OTX008 IC50 values for proliferation. (**A**) Subgroup of 35 up-regulated proteins and (**B**) subgroup of 17 down-regulated proteins in comparison to the protein profile in CAL 62 cells. Shades of red represent a high phosphorylation/expression profile in opposition to shades of blue corresponding to a weak phosphorylation/expression profile.

**Figure 6 cells-10-01112-f006:**
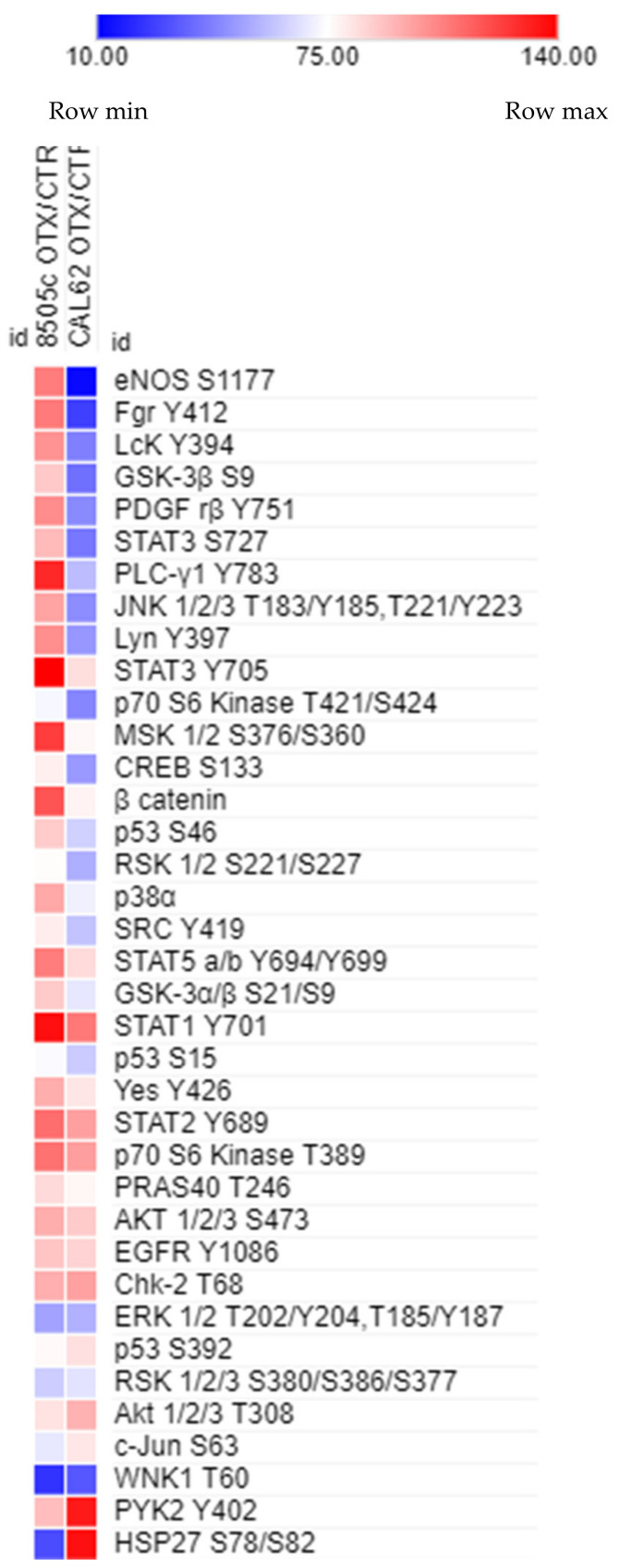
Phosphorylation profiles in anaplastic cell lines. Heatmap showing the phosphorylation level of 37 proteins according to the cell line (8505c vs. CAL 62) and the treatment (30 µM OTX008 30 min vs. CTR). Shades of red represent a high phosphorylation profile and shades of blue correspond to a weak phosphorylation expression.

**Figure 7 cells-10-01112-f007:**
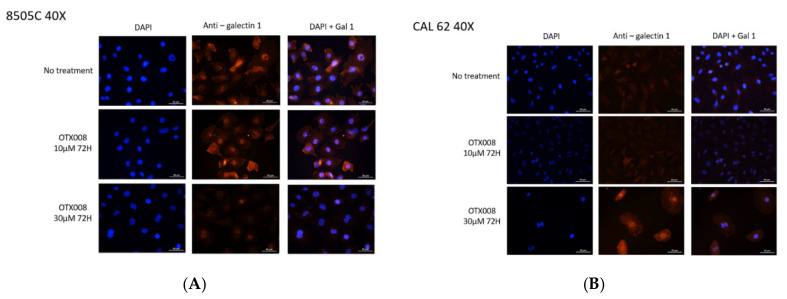
Immunofluorescence analyses using confocal analysis showing the expression of galectin-1 in anaplastic cell lines exposed to OTX008. Cell treatment with OTX008 affected cytoplasmic and nuclear localization in (**A**) 8505c cells and (**B**) CAL 62 cells.

**Figure 8 cells-10-01112-f008:**
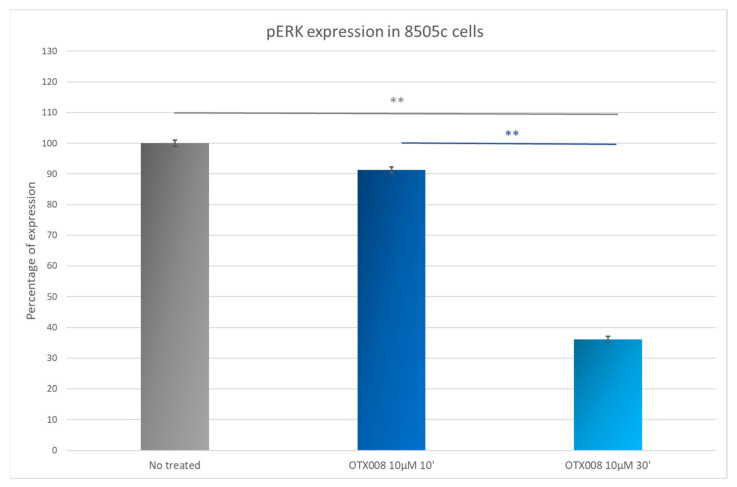
Evaluation of pERK level in 8505c cells. Cells were exposed for 10 and 30 min with 10 µM OTX008. ANOVA and Tukey post hoc tests were performed to assess significance ** *p* < 0.01).

**Figure 9 cells-10-01112-f009:**
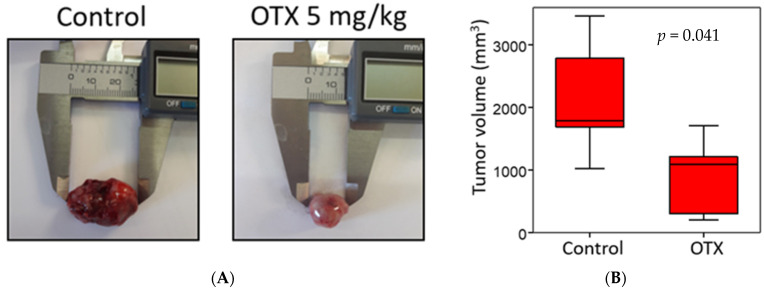
Effect of OTX008 on tumor volume developed from anaplastic cells in nude mice. (**A**) Illustration of two tumors and (**B**) box plot presenting tumor volume after mouse euthanasia in both control and OTX-treated groups. The Mann Whitney test was performed to assess significance.

**Figure 10 cells-10-01112-f010:**
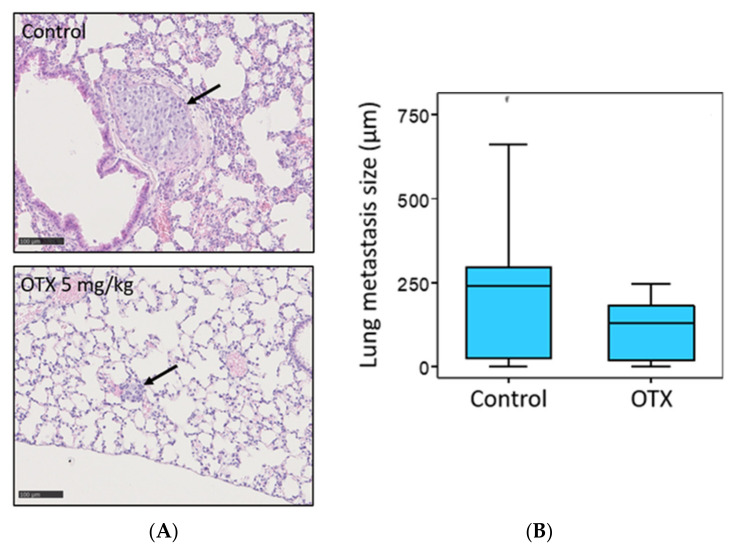
Evaluation of lung metastasis development from 8505c xenografts in control and OTX008-treated mice. (**A**) Lung sections colored by hematoxylin/eosine solution showing a large metastasis in a non-treated mouse and a small metastasis in a treated mouse. (**B**) Statistical comparison of lung metastasis sizes between control and OTX groups. The Mann Whitney test was conducted to evaluate significance.

**Table 1 cells-10-01112-t001:** Mutation status (expasy.org) of the six cell lines studied and the IC50 value (µM) for OTX008.

	Thyroid Cancer Type	Genomic Alterations												OTX008
		TERT	BRAF	NRAS	KRAS	p53	NF1	NF2	PTEN	CDK2	CDK6	RB1	RET	IC50 (µM)
BCPAP	Papillary	1	1	0	0	1	0	0	0	1	0	0	0	0.5
TPC-1	Papillary	1	0	0	0	0	0	0	0	0	0	0	1	0.2
FTC-133	Follicular	1	0	0	0	1	1	0	1	0	1	1	0	0.3
TT2609CO2	Follicular	1	0	1	0	1	0	0	0	0	0	0	0	1.1
8505c	Anaplastic	1	1	0	0	1	0	1	0	0	0	0	0	0.3
CAL-62	Anaplastic	0	0	0	1	1	0	1	0	0	0	0	0	30

## Data Availability

The datasets generated and/or analyzed during the current study are available from the corresponding authors on reasonable request.

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
