# Peer review of "New Treatment Strategy Targeting Galectin-1 against Thyroid Cancer"

_cells, 2021, doi:10.3390/cells10051112_

Round 1
Reviewer 1 Report
the authors show a long series of experiments on the activity of OTX008 in tumors expressing gal1 and its possible use in vivo.
1. if possible the authors could show an explanatory table of gal1 activity in the various cell lines
2. the author confirms the DL50 by MTT on the 8505c clarifies why in different experiments he uses different concentrations of OTX008 or uniforms the various experiments at the effective dose
3. the author compares the effects between inhibition by OTX008 vs silencing from a molecular point of view in 8505c
4. The author clarifies the lack of effect or lesser effect of 0TX008 on the CAL62 line (the different levels of phosphorylation and the role of eNOS?)
5. The author confirms the data of RPPA analysis on fresh tissues of patients.
6. Add a tumor growth curve of 8505c in vivo
7. Add liver and heart weight for lack of toxicity
8. Add the weight of the lung
Author Response
We thank the Reviewer for her/his great appreciation about our study.
- We added a brief conclusion in the “Abstract” of the article.
- NMR abbreviation is now replaced by the complete name in the “Introduction”.
- The sentence has been reworded to clearly describe the side effects we have examined, also in relation to the comments of Reviewer 2. In all treated mice, no adverse effects were found.
- The concentration of OTX008 for in vitro experiments was established from the effects of the drug on cell proliferation (Fig 1, crystal violet staining) and adapted in other experiments regarding to the time of exposure of the cells. The dose of OTX008 used in animals was based on the literature (Astorgues-Xerri L, Eur j Cancer 2014) and the amount of OTX008 was calculated regarding to the weight of the mouse (5 mg/kg/day) as described in “Material and Methods”.
Reviewer 2 Report
The author studied the role of (OTX008), galectin-1 inhibitor, to study its potential usage as a drug for Thyroid cancer using 3 types of Thyroid cancer for in-vitro studies and animal model. Since the author had already published Gal-1 role in thyroid cancer patients as well as the OTX008 have been used to treat other types of cancer, the novelty of the work declined.
However, there are major comments need to be addressed in the manuscript:
- It would be interesting to show the overexpression of Gal-1 in these cell-lines compared to normal ones. This might explain the variation of drug sensitivity.
- The author need to assess the toxicity of the drug using normal thyroid cells and compare it to the cancerous cell lines.
- The in-vitro studies lack the presence of primary cells which will impact the manuscript.
- The migration and invasion studies needs a negative control to compare with such as normal thyroid cells from same type.
- Author need to clarify what is the control in Fig3 and 4-B.
Author Response
We thank the Reviewer 2 for her/his comments helping us to improve our manuscript.
- The general Gal-1 activities are already developed in the “Introduction”, including cell-cell, cell-ECM interactions and regulation of intracellular signalling pathways. Specific activities of Gal-1 in our 6 lines are not known, but we compared its expression in 2 cell lines (8505c and CAL-62) and propose that the reduction of Gal-1 expression may be due to the KRAS mutation in CAL-62 cells and consequently resistance to OTX008. Also, our mechanistic study highlighted some key actors linked to the effects of OTX008 and therefore to the role of Gal-1.
- The IC50 values for OTX008 were calculated from crystal violet experiment (not MTT). In in vitro experiments, the concentrations of OTX008 have been adapted with the time of exposure of the cells to the drugs (10 or 30 min, 48 or 72 hours, or 10 days), in order to use effective concentrations. In many experiments, at least 2 concentrations were used, and our results reported concentration-dependent effects of OTX008, indicating specific activities. Of note, for mechanistic studies, (Figs 6-8), only the 2 anaplastic lines were examined, and we used a concentration as high as 30 µM which is the IC50 values for the resistant CAL-62 line.
- The Reviewer 2 proposed to compare the effect of OTX008 treatment and the Gal-1 silencing. Actually, one of our previous study already evaluated the effect of Gal-1 knockdown (KD) on thyroid cancer cells (Arcolia, Int J Oncol 2017). Indeed, our previous in vitro experiments revealed that migration was negatively affected in TPC-1 Gal-1 KD cells, and that proliferation and invasion capacity of 8505C cells decreased after Gal-1 KD. Moreover, an orthotopic mouse model displayed a lower rate of tumor development with galectin-1 KD thyroid anaplastic cancer cells than in the control. All Pof these data showed very similar effects between Gal-1 KD and OTX008, further supporting that Gal-1 is the mean target of the drug. This is in line with the knowledge that OTX008 binds to Gal-1 on the side backface, away from the b-galactoside-binding site (Dings, J Med Chem 2012) and that 3 µM OTX008 treatment induced a significant inhibition of in vitro invasion of SQ20B cells (a head and neck cancer line) after 48 h, while shLGALS1-RNA transfected cells had also lower invasion rates than control cells (Astorgues-Xerri, Eur J Cancer 2014).
- The weaker effect of OTX008 in CAL-62 line is due to the lower expression of Gal-1 in this cell line. Silencing Gal-1 in 8505c cells already inhibited cell migration for example and treatment of such KD cells with OTX008 will not bring additional information. Phosphorylation profiles also highlighted some pertinent regulations that we have discussed, and which could explain the resistance/sensitivity of the lines to the Gal-1 inhibitor.
- Fresh tissues from anaplastic thyroid cancer are not available at the Hospital “Institut Jules Bordet” because such disease is very rare. To validate RPPA data in patient samples, we investigate the cBioPortal database (Papillary Thyroid Carcinoma (TCGA, Firehose Legacy, 516 samples) (no RPPA available for anaplastic cancers) and examine co-expression of protein expression (RPPA) with Gal-1 mRNA, regarding our list of protein in Fig.5. This new analysis reports that Gal-1 positively correlates with ANXA1, LCK, YAP1, DVL3, EIF4EBP1, ERBB3, PTK2 and and negatively with AKT1, further supporting that the sensitivity to OTX008 correlates with protein regulation linked to Gal-1 expression.
- Please find below the tumor growth curve of 8505c in mice. This graph (fig 1) could be added in the manuscript if it is required by the Editor but it is somewhat redundant with the Fig.9B.
7-8. For in vivo experiments, lungs, livers and kidneys were collected and paraffin embedded. We did not observe significant effect of OTX008 on organ size or tissue morphology (see HE below) when mice were treated with OTX compared to vehicle. The only difference is the presence of smaller lung metastases in treated animals. We did not weight these organs at the time of sampling because organ weight is dependent of mouse weight. Altogether, in combination with no change in mouse weight and behavior during the treatment, our data indicate the absence of significant side effects when using 5mg/kg/day OTX008.
The figure 2 shows representative HE staining of lung, liver and kidney tissues of mice treated for 3 weeks with OXT008. No particularities were observable in the tissue and cell morphology in relation with the OTX treatment as validated by our Pathologist Dr Verset.
Reviewer 3 Report
The study by Gheysen et. al. is very well written and interesting to read. The findings are very well presented, especially the description of the laboratory proceedings. Only some minor commentaries need explanation in my opinion:
- Please add a short conclusion summarizing the results of your work to the abstract
- Introduction section - NMR analysis demonstrated that OTX008 targets – please add explanation to the abbreviation
- Section 3.9 - Of note, during treatment, no adverse effect, such as a significant reduction of animal weight or particular animal behavior, was observed. – Do the authors mean that there were no adverse effects whatsoever or that none of those two adverse effects were found? – if so, what adverse effects were found? Or taken into account?
- How was the necessary dosage of OTX008 calculated? What was the reason for the doses used for the research (dosage recommended by the producer?)
Author Response
On the contrary, it is because we have already published a research article showing the importance of the role of Gal-1 in thyroid cancer (Arcolia, Int J Oncol 2017) that the evaluation of the effect of a Gal-inhibitor 1 like OTX008 is of great importance. Moreover, as no results were found for OTX008 and thyroid cancer in PubMeg.gov, we confirm the originality of our study.
- As we published in 2017, normal thyroid cells did not express Gal-1 as demonstrated in patients by IHC (Arcolia, Int J Oncol 2017). Therefore, the evaluation of normal cells is completely clinically not relevant.
Moreover, the study from Astorgues-Xerri (Eur J Cancer 2014) already showed a very strong correlation between Gal-1 mRNA or protein expression and IC50 values for OTX008 (R2=0.58, p<0.0001) in a panel of human cancer cell lines.
Altogether, the relation between Gal-1 expression level and sensitivity of cell lines for OTX008 is already well-documented. This is now further reported in thyroid cancer with our current study.
- We assessed in our animal model the effect of OTX008 on normal cells not expressing Gal-1 such as in liver and kidney tissues. In these organs, we did not observe morphological changes. In relation with our above reply, we speculate that normal thyroid tissues should not be affected by OTX008. Unfortunately, thyroid glands were not collected in our study.
In addition, the paper from Chiariotti et al (Int J Cancer 1995) reported that Gal-1 mRNA and protein levels were higher in 6 thyroid carcinoma-derived cell lines than in normal thyroid primary cultures. Gal-1 mRNA levels increased in 28/40 papillary carcinomas and in 6/7 anaplastic carcinomas
compared with normal or hyperplastic thyroid. Immunohistochemical analysis of normal thyroid and papillary carcinoma sections revealed a higher content of Gal-1 protein in neoplastic follicular cells than in normal cells.
- Primary cells from anaplastic thyroid cancer are not available, even in collaboration with the Hospital “Institut Jules Bordet” because such disease is very rare. So, we prefer to work with a panel of well-established and well-characterized thyroid cancer cell lines.
- Migration and invasion of normal cells are not the aims of our study. As commonly used in the literature, negative controls are cancer cells with no treatment. On the other hand, normal cells always need a specific medium with many stimulators of cell survival for culture, so in these conditions it is not acceptable to use these additional controls for comparison with cancer cells.
- We agree with the final comment of the Reviewer 3 and bring additional details about the control. Actually, migration and invasion of untreated cells are normalized at 100% for all lines and it is why we represented only one control bar to simplify the figure. This is now more explain in legend figures 3 and 4.
Round 2
Reviewer 1 Report
I think in this form the paper i suitable for pubblication!
Author Response
We thank the Reviewers who appreciate our replies and changes.
Reviewer 2 Report
I want to thank the author for clarifying my comments but even though this publication should not be dependent on previous publication. I think expression profile of Gal1 should be reassessed to impact this paper.
Author Response
We thank the Reviewers who appreciate our replies and changes.
To further improve the quality of the current manuscript and as suggested by the second Reviewer, we added Supplementary Data reporting the Gal-1 expression profile in the 6 thyroid cancer lines. These data are now described in the section 3.7. We agree that these results are usable as Gal-1 is the target for OTX008, and we hope that like this, our study is suitable for publication in the “Cells” journal.